# Lite and Efficient Deep Learning Model for Bearing Fault Diagnosis Using the CWRU Dataset

**DOI:** 10.3390/s23063157

**Published:** 2023-03-15

**Authors:** Yubin Yoo, Hangyeol Jo, Sang-Woo Ban

**Affiliations:** 1Department of Information & Communication Engineering, Graduate School, Dongguk University, Gyeongju 38066, Republic of Korea; 2Department of Electronics, Information & Communication Engineering, Dongguk University, Gyeongju 38066, Republic of Korea

**Keywords:** bearing fault diagnosis, convolutional neural networks, spectrogram, short-time Fourier transform, CWRU dataset

## Abstract

Bearing defects are a common problem in rotating machines and equipment that can lead to unexpected downtime, costly repairs, and even safety hazards. Diagnosing bearing defects is crucial for preventative maintenance, and deep learning models have shown promising results in this field. On the other hand, the high complexity of these models can lead to high computational and data processing costs, making their practical implementation challenging. Recent studies have focused on optimizing these models by reducing their size and complexity, but these methods often compromise classification performance. This paper proposes a new approach that reduces the dimensionality of input data and optimizes the model structure simultaneously. A much lower input data dimension than that of existing deep learning models was achieved by downsampling the vibration sensor signals used for bearing defect diagnosis and constructing spectrograms. This paper introduces a lite convolutional neural network (CNN) model with fixed feature map dimensions that achieve high classification accuracy with low-dimensional input data. The vibration sensor signals used for bearing defect diagnosis were first downsampled to reduce the dimensionality of the input data. Next, spectrograms were constructed using the signals of the minimum interval. Experiments were conducted using the vibration sensor signals from the Case Western Reserve University (CWRU) dataset. The experimental results show that the proposed method could be highly efficient in terms of computation while maintaining outstanding classification performance. The results show that the proposed method outperformed a state-of-the-art model for bearing defect diagnosis under different conditions. This approach is not limited to the field of bearing failure diagnosis, but could be applied potentially to other fields that require the analysis of high-dimensional time series data.

## 1. Introduction

Considerable research has been conducted to develop variant and powerful deep learning models which are also applied to various applications, and have contributed significantly to developing many innovative, intelligent systems. Deep learning model research is being actively conducted, mainly on image and voice understanding and recognition, which has expanded to various application fields. In particular, remarkable research results have been obtained in natural scene understanding, natural language understanding, and human voice recognition. These developed models are being utilized in the medical field, such as in diagnosing various diseases and diagnosing machine failures in industry. This machine fault diagnosis function is making a significant contribution to the smart factory in pursuit of the high efficiency of a group of connected machines. Among the various machine faults, bearing faults are very common and critical. Several surveys of induction machine failures have revealed bearing failures to be the most common, accounting for 30–40% of all machine failures [1,2,3]. 

In particular, in wind power generation, bearing fault diagnosis is becoming a critical issue [4]. Recently, the construction of wind power plants has been expanding to help achieve carbon neutrality and ensure safety from the dangers of nuclear energy. Accordingly, in addition to strengthening the predictability of wind power generation for the stable electricity production of wind power stations, the health monitoring of wind power generators is becoming an important issue [5]. 

In recent years, considerable research has been conducted on utilizing deep learning models for bearing failure diagnosis, with numerous studies published since 2016 [6,7,8,9,10,11,12,13,14]. These studies demonstrated the remarkable potential of deep learning models in accurately diagnosing bearing defects using vibration sensor signals. 

On the other hand, many studies have been conducted to improve the performance of bearing failure diagnosis using more complex models applied to image or voice understanding. Recently, research for developing lower complex models has been conducted for failure diagnosis, using systems with limited hardware specifications such as edge computing. In addition, research is being conducted to reduce the complexity of high-complexity models with excellent classification performance by applying optimization techniques [15]. In many studies using lightweight models, however, the classification performance is inferior to that of models with high complexity.

Therefore, this paper proposes a bearing failure diagnosis model that can reduce the model complexity and computation time significantly while maintaining excellent classification performance by applying a low-complexity CNN model and low-dimensional input data.

When an abnormality occurs in a specific part of the bearing, the characteristics of the bearing vibration sensor signal are different from the vibration pattern in the normal situation, which alters the frequency characteristic of the signal. Considering these characteristics, the spectrogram is used as input data for the proposed deep learning model. The spectrogram reflects the time–frequency features of signals and is generated through a short-time Fourier transform of vibration sensor signal. 

In this study, the vibration sensor signal was downsampled, and a small-sized spectrogram was constructed using the minimum interval. Furthermore, a model optimized for low input data dimensions was designed by fixing the feature map dimensions. The effectiveness of the proposed approach was verified by conducting experiments using the well-known and widely used Case Western Reserve University (CWRU) dataset [16], which contains various bearing fault conditions, including inner race, outer race, and roller defects, as well as different motor speeds and load conditions. The experimental results showed that the approach could reduce the computational and memory requirements significantly while maintaining high diagnostic accuracy for bearing defects under all conditions.

When applied to actual industrial sites, bearing failures can be diagnosed with little computing power. The sensor cost can be reduced using a sensor with low sample rate characteristics in the data acquisition stage. This can be important for machines where it can be critical to identify faults quickly and take plausible action on machine failures.

The remainder of the paper is outlined as follows. Section 2 introduces various models related to bearing defect diagnosis. Section 3 describes the proposed bearing defect diagnosis model and techniques for the dimensionality reduction of input data. The experimental results of the proposed model are reported in Section 4. Finally, the conclusion and further works are outlined in Section 5.

## 2. Related Works

Even if it is limited only to the bearing failure diagnosis problem, considerable research has been conducted to solve this diagnosis problem. Research related to bearing defect diagnosis has mainly focused on developing systems that acquire bearing condition signals from vibration sensors and automatically diagnose bearing defects by applying various machine learning techniques to the acquired vibration sensor signals [17]. Early studies applying machine learning methods used a multilayer perceptron (MLP) and a support vector machine (SVM) using the original signal measured from the vibration sensor or the frequency characteristic information of the vibration sensor signal. As a result of this study, the multilayer perceptron model showed 80–90% defect diagnosis performance [18], and the defect diagnosis using the support vector machine showed a maximum of 98.5% [19]. Multilayer perceptron and support vector machines have various feature extraction techniques from sensor signals to improve their performance, but they have limitations in their defect diagnosis performance, which lacks robustness to changes in input signal characteristics.

Research on implementing a bearing defect diagnosis system using a deep learning model has been actively conducted to improve on these problems. Owing to the nature of the deep learning model, both feature extraction and defect classification became possible through learning without applying a special feature extraction technique in advance. Chen et al. [6]. converted the spectrogram obtained from the vibration sensor signal of the CWRU bearing dataset into a size of 224 × 224 and used it for learning. Chen et al. [6]. proposed a model for classifying bearing defects by applying transfer learning techniques to the ResNet-50 model applied to the image classification of ImageNet. The classification performance of the proposed model ranged from 99.90% to 100%. Regarding bearing defect diagnosis performance, it is an excellent model compared to models applied with previous machine learning techniques [6]. Deveci et al. [7]. proposed a study in which preprocessing was performed by adding white Gaussian noise to consider the noise characteristics of the vibration sensor of the CWRU bearing dataset.

Deveci et al. used AlexNet, GoogLeNet, and ResNet-50 models to diagnose bearing defects, with 224 × 224 and 227 × 227 spectrogram data as the input data of the deep learning model [7]. For each applied deep learning model, the classification performances of the AlexNet, GoogLeNet, and ResNet-50 models were 97.08%, 97.60%, and 99.27%, respectively [7]. Thus, the best bearing defect classification performance was shown when ResNet-50 was applied. In the case of the ResNet-50 model, however, high computational complexity is required because the number of parameters to be learned is high, so high-end hardware is required to develop a defect diagnosis system capable of real-time processing in an actual industrial site. Building a high-end, expensive system for machine defect diagnosis can be an economic burden. In the case of building a low-end, low-cost system, there is a limitation in prompt defect diagnosis. When a machine needs to be stopped urgently according to defect diagnosis, it can adversely affect the productivity of the machine.

Research on bearing defect diagnosis based on a CNN, rather than complex models such as the ResNet-50 model and GoogLeNet model, is being actively conducted. Neupane et al. proposed a CNN model for diagnosing bearing defects using 224 × 224 input data by converting the CWRU bearing dataset from vibration sensor signals into scalograms and spectrograms. The bearing defect classification performance of the CNN-based model proposed by Neupane et al. is 99.88% [1], which is superior to the model using the multilayer perceptron and support vector machine. On the other hand, the bearing defect classification performance is slightly lower than that of the model using ResNet-50 [1].

In addition, research on fault diagnosis using a lite CNN model with reduced complexity is being actively conducted to implement a fault diagnosis system using a system with low hardware specifications. The lite CNN model proposed by Mukherjee et al. [15]. has a simple structure, low computation, and small model size compared to existing models with high complexity [15]. Nevertheless, it has a large input data dimension, and its classification performance decreases significantly by ~50% under certain conditions.

Directly comparing the superiority of models based on the performance results alone may be inappropriate because there are obvious limitations to comparing these models under the same conditions. However, in a study using the CWRU dataset, the Chen et al. [6] model showed the best performance regarding the quantitative defect diagnosis performance. Therefore, this paper attempts to show that the proposed model generates superior performance compared to the model proposed by Chen et al. [6].

## 3. Proposed Bearing Fault Diagnosis Model

The proposed bearing defect diagnosis model in this work is a modified CNN model, and the training data of the proposed model is the spectrogram obtained from short-time Fourier transform (STFT) after downsampling the vibration sensor signal of a 12K/48K sample rate to 6K. Figure 1 presents the bearing failure diagnosis of the proposed light CNN model.

ResNet-based models show high classification accuracy in bearing failure diagnosis but require a large amount of computation because of the large number of convolutional layer structures and large dimensions of input data. Lite CNN-based models have a small amount of computation owing to their low-complexity structure but have low classification performance. With much lower complexity and less computation time, the proposed model produces comparable performance to the ResNet-based state-of-art model.

### 3.1. Lite Convolutional Neural Network Model (Lite CNN)

The lite CNN model proposed for bearing defect diagnosis consists of a convolution layer and a fully-connected layer. Figure 2 shows the overall structure of the proposed model.

The convolutional layer applies 64 kernels, 3 × 3 in size, and 32 kernels, 1 × 1 in size, to improve the classification performance. Generally, a pooling layer is placed after the convolutional layer to reduce the computational cost by reducing the dimensions of the feature map and the number of network parameters. In the proposed model, a pooling layer was not applied to maintain the dimension of the input data. A convolution layer applies a padding technique to maintain the size of the feature map of the hidden network as the same size as the input data. The fully connected layer serves to classify the feature map generated in the final feature extraction layer. The fully connected layer has two hidden layers with 256 nodes for acceptable classification performance. Overfitting was prevented by applying a dropout between fully connected layers, and a softmax function was used as an activation function of the output layer. Adam was applied as an objective function optimizer to improve classification performance.

### 3.2. Dimensionality Reduction of Input Data

The proposed lite convolutional neural network (CNN) model has a compact and lightweight structure designed to achieve high classification performance comparable to ResNet-based models with higher complexity. Moreover, further reductions of the complexity of the proposed lite CNN model have been considered by reducing the dimensionality of the input data. 

The CWRU dataset consists of vibration sensor data obtained by inducing damage to the bearings of a rotating machine. Typically, damage to the bearings of rotating machines results in periodic signals corresponding to one revolution of the drive shaft of a facility. The CWRU dataset was recorded from a facility rotating at an average of 1750 RPM, and the vibration sensor signals corresponding to 1 revolution are approximately 0.0344 s. The vibration sensor signals have been transformed into spectrograms in the time–frequency domain using the short-time Fourier transform (STFT) to generate the input data of the proposed lite CNN model. The model was configured to diagnose the bearing fault using the shortest sections of the signals, similar to 0.0344 s, reducing the dimension of the input data.

Furthermore, the dimensionality of the input data is reduced by adjusting the sample rate of the vibration sensor signals. The CWRU dataset includes vibration signals measured at high sample rates of 12K and 48K. This study investigated the optimal sample rate that can achieve high classification performance for bearing failure diagnosis even with signals having sample rates lower than 12K and 48K. Entropy change analysis of the input spectrogram data was considered to analyze the adequacy of the degree of dimensionality reduction applied to the proposed lite CNN model. The image entropy analysis technique was used to analyze the entropy of the spectrogram.

Image entropy analysis is a technique used to measure the randomness or uncertainty of an image. It quantitatively measures the amount of information or entropy present in an image. The entropy of an image is calculated using Equation (1):(1)$$H=-\sum p_{i}\log_{2}p_{i}$$
where $p_{i}$ is the probability of the *i*th color level in the image. The color levels in an image represent the intensity values of the pixels, and their probabilities are calculated based on the number of pixels having each intensity value. 

The procedure of spectrogram entropy analysis is as follows:Divide the pixel values of the spectrogram into *n* bins.Calculate the histogram of the spectrogram using *n* bins, which provide the number of pixels with each gray level interval in its corresponding bin.Normalize the histogram by dividing each bin count by the total number of pixels to obtain the probability distribution.Calculate the entropy of the spectrogram using Equation (1).

In the case of a bearing failure, certain frequencies are strongly expressed, known as fault frequencies. For ball bearings, the ball pass frequency of the outer race (*BPFO*), ball pass frequency of the inner race (*BPFI*), and ball spin frequency (*BSF*), depending on the fault area, are the frequently occurring fault frequencies. These fault frequencies represent damage to the outer ring, inner ring, and ball area of the bearing, respectively. The fault frequencies can be calculated using Equations (2)–(4) [20]:(2)$$BPFO=\frac{N}{2}f_{o}-f_{i}\left(1+\frac{B}{P}\cos\phi \right)$$
(3)$$BPFI=\frac{N}{2}f_{o}-f_{i}\left(1-\frac{B}{P}\cos\phi \right)$$
(4)$$BSF=\frac{N}{2B}f_{o}-f_{i}\left[\left(1-{\left(\frac{B}{P}\right)}^{2}{cos}^{2}\phi \right)\right]$$
where $f_{i}$ is the frequency of inner race rotation; $f_{o}$ is the frequency of outer race rotation; N is the number of balls; ϕ is the contact angle; B is the diameter of balls; and P is the pitch diameter of the basic load.

The fault frequency of a ball-bearing component is typically expressed as an impulse signal in the vibration sensor signal when the component is damaged. Figure 3 presents a spectrogram generated by the vibration sensor signal for a bearing with a faulty outer race and a bearing under normal conditions. 

When generating a spectrogram in the time–frequency domain, the fault frequency component is expressed strongly in the spectrogram of the defective bearing. Normal bearings do not exhibit fault frequencies and generally have a flat spectrogram value. Normal bearings have high certainty with low information, while bearings with defects produce a spectrogram with relatively high information and low uncertainty because of the fault frequency.

On the other hand, entropy analysis has difficulties in determining the various parameters used in the analysis [21,22]. Various signal analyses are required for parameter optimization to efficiently classify bearing defects using entropy analysis [21]. Additionally, entropy analysis has poorer overall performance than fault diagnosis using deep learning.

Therefore, in this study, the defect frequency component of the bearing was used as an index to compare the amount of information expressed in the spectrogram. When downsampling the vibration sensor signal to generate the spectrogram, it was used limitedly to select the optimal downsampling candidate that maintained a strong expression of the defect frequency. The vibration sensor signal was downsampled using the candidate regions obtained through entropy analysis. A spectrogram was generated from the downsampled vibration sensor signal and used as input data for the proposed lite CNN model for experiments.

Downsampling the 12K and 48K sample rate signals to 6K can reduce the input data by half and reduce the dimensionality of the data by half when generating a spectrogram. Figure 4 illustrates a downsampled vibration sensor signal and the corresponding spectrogram.

The size of the spectrogram converted from the vibration sensor signal downsampled to the proposed 6K was reduced by 50% on the vertical axis compared to the original signal with a sample rate of 12K. The method of obtaining effective signal values every N times is used during the downsampling process instead of relying on the high sample rate signal data. This results in a significant loss of signal information. On the other hand, it still enables the construction of data with a signal similar to that measured by a sensor with an actual low sample rate. This makes it possible to use a vibration sensor signal with a low sample rate characteristic for diagnosing bearing failures in actual facilities, thereby reducing the cost of acquiring data used for this purpose. This is a critical issue in the diagnosis of the conditions of complex equipment that requires the use of multiple sensors.

In this study, the input data dimension was kept, small using the spectrogram as input data for a short duration and utilizing the downsampled vibration sensor signal. Typically, ResNet-based models require a fixed size of 224 × 224 or large inputs due to multiple dimensionality reductions that occur during numerous convolution operations. On the other hand, using a lite CNN model that is designed to use a smaller feature map size, it is possible to maintain high classification performance while still utilizing relatively small input data.

### 3.3. Spectrogram Using Short-Time Fourier Transform (STFT)

A vibration sensor measures the acceleration, velocity, or displacement of a machine component, and converts this mechanical motion into an electrical signal that can be analyzed. The signal typically contains information about the frequency, amplitude, and phase of the vibration.

After converting the vibration sensor signals in the CWRU dataset into spectrograms using the short-time Fourier transform (STFT) operation, they were input into the proposed model. The use of spectrograms in the time–frequency domain obtained through STFT transformation is a common technique because it allows for an effective observation of the time–frequency characteristics of time series data. 

STFT is a time–frequency analysis technique that decomposes a signal into its frequency components by applying the Fourier transform to overlapping signal segments. The basic idea behind the STFT is to divide a time domain signal into multiple overlapping segments, also known as windows, and then perform the Fourier transform on each window to obtain the frequency information. The frequency information of each window is then represented as a spectrogram, which is a two-dimensional representation of the signal in the time–frequency domain.

Formally, the STFT of a signal x(t) with a window function w(t) can be represented using Equation (5):(5)$$X(\omega ,t)=\int x(\tau )w(\tau -t)e^{-j\omega \tau }d\tau$$
where ω is the frequency and *t* is the time. The choice of the window function can affect the resolution of the spectrogram in the time and frequency domains. The resulting spectrogram can then be visualized as an image, where each row represents a frequency component, and each column represents a time window. The intensity of each pixel in the spectrogram represents the magnitude of the frequency component at a particular time window [23].

Figure 5 shows the process of transforming the vibration sensor signals into spectrograms in the time–frequency domain through STFT.

The vibration sensor signal is divided into sections with a hop size equal to the window size, and a Fourier transform is continuously applied. This study determined the optimal spectrogram generation parameters experimentally based on the characteristics of the vibration sensor signal in the CWRU dataset. Figure 6 shows the vibration sensor signal recorded by the drive end sensor for a 12K drive end bearing defect. Figure 7 presents the converted spectrograms of the corresponding vibration sensor signals in Figure 6. As shown in Figure 7, the characteristics of the type and size of the defects were more clearly distinguishable in the time–frequency domain spectrogram form than in the original vibration sensor signal shown in Figure 6.

### 3.4. Evaluation Metrics

Accuracy, parameter, floating-point operations (*FLOPs*), and computation time as evaluation metrics were used. Accuracy is an evaluation metric widely used to assess the classification performance of deep learning models. The metric represents the ratio of correctly classified samples to the total number of samples. The equation for calculating accuracy is expressed as Equation (6):(6)$$Accuracy=\frac{TP+TN}{TP+TN+FP+FN}$$
where *TP, TN, FP,* and *FN* refer to the true positive, true negative, false positive, and false negative, respectively. Evaluation metrics, such as the number of parameters and *FLOPs*, were used to compare the complexity of deep learning models. The number of parameters represents the weights, node count, and dimensions of input and output data of the deep learning model. *FLOPs* indicate the number of matrix operations performed repeatedly for computation in the deep learning model. Computation time is the sum of the training and classification times of the model. The formulae for calculating the number of parameters and *FLOPs* are expressed as Equations (7) and (8):(7)$$Parameters=k*k*c_{i-1}*c_{i}$$
(8)$$FLOPs=k*k*m_{i}*m_{i}*c_{i-1}*c_{i}$$
where k is the filter size; $m_{i}$ is the current feature map size; $c_{i}$ is the number of filters in the current layer; and $c_{i-1}$ is the number of filters in the previous layer. The total number of parameters and *FLOPs* are calculated by summing up the number of parameters and *FLOPs* computed at each layer. The classification performance of the proposed model was evaluated using Accuracy, and its complexity was compared using the Parameter and *FLOP* evaluation metrics. Computation time refers to the total time required for the training and classification of a deep learning model.

## 4. Experimental Results

In this study, the CWRU dataset was used to evaluate the classification performance and complexity of the proposed lite CNN model using low-dimensional input data. The classification accuracy and complexity of the proposed model were compared with various deep learning models of different complexities. Experiments were conducted by reducing the dimensionality of the input data using the proposed model. Python 3.7, with libraries such as Tensorflow, Keras, Scikit-learn, and Librosa, was used to implement the proposed framework. The computer used in this study was equipped with an Intel i7-11700K processor, 128GB of memory, and an NVIDIA GeForce RTX 3080Ti graphics card, running on the Windows 10 operating system.

### 4.1. Dataset Description

The CWRU bearing dataset is one of the most widely utilized datasets in bearing research, made available by the Bearing Data Center at Case Western Reserve University [16]. It features vibration sensor data for normal and defective bearings, including normal, drive-damaged, and fan-damaged bearings, collected using vibration sensors. The CWRU Bearing dataset is commonly used to evaluate and compare the performance of bearing defect diagnosis models [23].

Figure 8 presents the experimental setup used to obtain the CWRU bearing dataset. The setup consists of a two horsepower motor, token converter, encoder, dynamometer, and control device, which are assembled in a bearing test device. A typical bearing consists of an inner race, an outer race, balls, and a cage that holds the balls in place. The bearing supports the motor shaft in the test setup, and torque is transmitted to the shaft through the dynamometer and electronic control system.

The CWRU bearing dataset collects vibration sensor signals for normal and defective bearings. The vibration sensors are located at three different places: the drive end, fan end, and basement of the experimental setup. The sensors were classified into BA, DE, and FE based on their location. The artificial defects in the test bearings were divided into three types: inner race, outer race, and ball, with defect diameters of 0.007″, 0.014″, 0.021″, and 0.028″. The outer race was measured in three directions (3 o’clock (90°), 6 o’clock (180°), and 12 o’clock (0°)) based on the load direction. The vibration sensor data was collected under different motor speeds (1797, 1772, 1750, and 1730 rpm) and motor loads (0, 1, 2, and 3 horsepower) based on the damage size and site. The drive end bearing defect data was collected with two samplers (12 KHz and 48 KHz), and the fan end bearing defect data were collected only at 12 KHz. Normal data were collected only at 48 KHz.

### 4.2. Fault Diagnosis Using the Proposed Model (Lite CNN)

This study compared the performance of various deep learning models with different levels of complexity. The performance of ResNet50 was compared with the proposed lite CNN with adjusted convolution layers—CNN0, CNN2, and CNN3—to determine the optimal lightweight model. Figure 9 shows the structures of the models used in the comparison.

This study conducted experiments using the vibration sensor signals in normal conditions measured at 12K and 48K sample rates, and vibration sensor signals measured at the drive end sensor in the case of drive end bearing faults. The faults were classified into 12 classes based on the type of damage to the bearings. This study used 65 × 50 spectrograms as input data for the comparison models. Table 1 lists the parameters used to generate the 65 × 50 spectrograms. Different segmentations, windows, and hop sizes were applied to generate spectrograms of the same size using different raw vibration signals.

Figure 10 shows examples of spectrograms generated using the experimental parameters.

Table 2 lists the results of ten consecutive experiments. Three hundred training data, 100 validation data, and 200 test data were used for each class, resulting in 3600 training data, 1200 validation data, and 2400 test data for 12 classes. The proposed lite CNN model showed the best classification performance and required a short computation time owing to its low complexity. The proposed model has a similar total number of parameters to the CNN0 model without a convolution layer and has fewer parameters than other CNN-based comparison models. In addition, it achieved superior classification performance with only 2.465% of the computation cost compared to ResNet50.

### 4.3. Fault Diagnosis Using the Less Input Dimension

Spectrograms were generated using the minimum length of the vibration sensor signal and downsampling to reduce the dimensionality of the input data used in the lite CNN model. The generated spectrograms were used as input data for the lite CNN model in the experiments. 

The optimal signal length that maintains the classification performance of the lite CNN model was determined by constructing spectrograms using four signal lengths. The vibration sensor signals measured at 12K and 48K sample rates were used for drive end bearing damage from sensors located at the drive end. Table 3 lists the parameters used to construct the spectrograms for the four signal lengths used in the experiment, with the parameters in parentheses used for the 48K signal.

Figure 11 shows examples of spectrograms constructed using four different signal lengths. The same window and hop size were used to maintain consistent time and frequency intervals, while the segmentation size was adjusted to vary the size of the spectrograms. 

Table 4 lists the results of 10 consecutive repeat experiments. A total of 300 train data, 100 validation data, and 200 test data were used for each class, resulting in 3600 train data, 1200 validation data, and 2400 test data for 12 classes. The 65 × 10-sized spectrogram generated using a signal interval of 0.0346 s maintained excellent classification performance despite its small size, indicating the appropriateness of using vibration sensor signals corresponding to a single rotation cycle of the rotating machine for bearing fault diagnosis. In addition, the proposed method demonstrated the potential to further reduce the model parameters, computational complexity, and operation time. 

Figure 12 represents the classification accuracy derived from the experiments. The classification performance of the 65 × 5 size spectrogram generated using a signal segment of 0.0213 s, which is shorter than 1 rotation cycle of the machine, decreased significantly.

The lite CNN, with a lightweight structure, was used to determine the minimum length of the vibration sensor signal required for bearing fault diagnosis. The vibration sensor signal was downsampled to reduce the dimensionality of the input data further. The vibration sensor signals measured at 12K and 48K sample rates from the sensor located at the drive end were used to diagnose bearing damage in the drive end. Table 5 lists the parameters used to downsample the vibration sensor signal and transform it into a spectrogram. The time–frequency domain spectrograms were constructed to have the same time interval by adjusting the segmentation, window, and hop size. The constructed spectrograms showed differences in the interval of the frequency domain.

Figure 13 presents examples of spectrograms generated using the experimental parameters.

Entropy analysis was performed using a spectrogram generated from vibration sensor signals to select the optimal downsampling candidate. Fifteen arbitrary segments were divided, and the spectrogram values corresponding to each segment were measured for entropy analysis. This analysis allows for a relative comparison of the information content of frequency components that strongly manifest when bearing faults occur. Figure 14 shows the results of entropy analysis performed on spectrograms of various sizes generated from vibration sensor signals measured on a 0.007 diameter damage in the ball region. As the size of the spectrogram decreased due to downsampling, the entropy value increased. In particular, the entropy increased significantly in the 16 × 10 spectrogram generated by downsampling at a 3K sample rate. Therefore, the 32 × 10 spectrogram generated by downsampling at a 6K sample rate might be suitable as the input for the lite CNN model for diagnosing bearing faults.

Experiments were conducted using the lite CNN model to confirm the suitability of the downsampling interval estimated by entropy analysis. Table 6 lists the experimental results, which indicate that the estimated sampling rate of 6K is appropriate. The 32 × 10 spectrogram generated by downsampling to 6K maintains excellent performance in classification accuracy compared to signals with higher sampling rates, with almost no difference. The 16 × 10 spectrogram generated by downsampling to 3K, which is similar to the results of the entropy analysis, reduced the classification accuracy significantly. Furthermore, the experiments showed that downsampling the vibration sensor signal can effectively reduce the computational complexity and parameter count of the deep learning model.

### 4.4. Comparison Experiment with the Best-Performing Model

Bearing failures were diagnosed while significantly reducing the complexity of the model using a dimensionality reduction technique for input data based on the structure of the lite CNN model and the characteristics of vibration sensor signals.

Table 7 lists the performance of the proposed technique with the transfer learning-based ResNet50 model [6] using the same CWRU dataset. The same number of training, validation, and test data were used, and the proposed lite CNN model was tested with 10 classification classes under the same conditions. To produce 10 classes, data with a 0.028 diameter inner race and ball damage were excluded. For each class, 300 train data, 100 validation data, and 200 test data were used. Three thousand train data, 1000 validation data, and 2000 test data were used for the 10 classes.

The comparison was made by considering the calculation amount and the total number of parameters, excluding the dense layer, of the transfer learning-based ResNet50 model because of the structural difference from the lite CNN model. This was done for a more accurate comparison.

Compared to the transfer learning-based ResNet50 model, the proposed model showed the same average classification accuracy and maximum classification performance, which was 0.02% higher in the 10 repeated experiments. Despite the low complexity, the proposed model produced better classification results than the SOTA model [6]. On the other hand, both models have excellent classification properties, considering the experimental conditions.

The proposed lite CNN model using 32 × 10 input data dimensions significantly reduces the computation complexity in terms of the total number of parameters, FLOPs, and computation time compared to the SOTA model. The lite CNN model had only 0.64% of the total number of parameters, 0.05% of the total FLOPs, and 6.399% of the computation time compared to the SOTA model. This results in a significantly lighter model with the same maximum classification performance. Under a limited computing power environment, the proposed lite CNN model’s 32 × 10 input data dimension was plausible and ideal for fault diagnosis [15].

### 4.5. Generalization Performance Experiment

The proposed lite CNN model exhibited excellent classification performance, even with a lightweight structure, under the experimental conditions of Set1 of the CWRU dataset.

The CWRU dataset was composed of three different types of defects located in two areas of a rotating machine (drive end and fan end) and measured by three different sensors (drive end, fan end, and base) at different locations. The dataset contained a combination of vibration signals in the time domain. The proposed lite CNN model was tested on the data from all sensor types to classify the generalized characteristics of bearing failures in the facility. Table 8 lists the composition of the CWRU dataset used in the experiments.

Table 9 presents the experimental results of the proposed model using the vibration sensor signals measured at various damaged locations and sensor positions. For each class, 300 train data, 100 validation data, and 200 test data were used.

The classification experiments performed on the CWRU dataset using the drive end, fan end, and base sensors for all types of damage confirmed that the proposed lite CNN model had excellent classification performance, similar to that obtained when using the drive end sensor to classify drive end damage under the experimental conditions of Set 1. In the experimental conditions of Sets 2 to 8, a high classification accuracy of at least 99.65% or more was confirmed in 10 repeated experiments. All experimental data achieved a high average classification performance of 99.78 to 99.98%.

Hence, the proposed lite CNN model generates excellent classification results for all data sets of the CWRU.

## 5. Conclusions and Further Works

This paper proposes a new lightweight CNN model for bearing defect diagnosis based on low-dimensional input data that considers equipment and signal characteristics. The model achieved competitive diagnostic performance while having low complexity. Two methods were proposed to reduce the dimensionality of the data required for bearing fault diagnosis. First, an effective 0.0346 s signal segment for bearing fault diagnosis was identified by calculating the rotational cycle of the rotating machinery. Second, the vibration sensor signals obtained for bearing faults were downsampled to 6K by reducing the sampling rate. The spectrograms generated using the proposed techniques had lower input data dimensions than the 224 × 224 input data dimensions used by the transfer learning-based ResNet50 model. To detect bearing faults using low-dimensional input data, a lite CNN model was proposed that uses padding and excludes pooling to prevent a reduction in the feature map dimensions during convolution and pooling operations. The experiments performed using the CWRU dataset showed that the proposed model achieved an average of 0.02% superior classification performance with only 0.64% total parameters, 0.05% total FLOPs, and 6.399% computation time compared to the latest models, including the SOTA model. Furthermore, the proposed model could effectively classify various bearing failures, regardless of the location or type of sensor used to measure the data in the same facility.

One limitation of the proposed light deep learning model for bearing fault diagnosis is that it might not capture all the complexity and variability of the data. The light models, like the proposed model, typically have fewer layers and parameters, which can reduce their ability to learn and generalize from the data. In addition, reducing the input data dimension may result in a loss of important information that could affect the accuracy of the diagnosis. Therefore, it is essential to evaluate the trade-off between computational efficiency and classification performance carefully when using light deep learning models for bearing fault diagnosis. Another limitation is that the proposed model may not be suitable for detecting rare or subtle faults that require more complex and sophisticated models to capture their patterns. Furthermore, low sample rate sensors may lead to a loss of information critical for identifying specific bearing defects. The effectiveness of the light model may also depend on the specific operating conditions and characteristics of the rotating machinery being analyzed. Therefore, while the proposed light deep learning model offers a practical solution for reducing computational and data processing costs, it might not always be the best choice for all applications of bearing fault diagnosis.

When diagnosing faults in complex systems, it is vital to consider the complexity of the problems that may arise. One way to do this is to develop models that capture the intricacies of a system appropriately, without adding unnecessary complexity that could lead to overfitting or computational inefficiencies. Hence, it is crucial to gather diverse data from real-world systems in operation, which can be used to test and refine the models. By utilizing such data, lightweight models can be developed with enhanced generalization characteristics that are better suited for fault diagnosis. Conducting experiments with this data can help ensure that the proposed model is accurate, reliable, and efficient, and can be applied to various systems in various industrial fields. Ultimately, this research aimed to improve fault diagnosis and reduce downtime and maintenance costs in complex systems, leading to more efficient and reliable operations.

On the other hand, the proposed method has only been tested on the CWRU dataset. While it showed effective results for diagnosing bearing faults in the specific dataset, it cannot be applied uniformly to all rotating machinery, operating conditions, and bearings. Therefore, the length and sample rate of vibration sensor signals should be recalculated and adjusted for each dataset when testing under different conditions.

Future research will validate the proposed method using various types of bearing fault data with noise from real operating machinery. More research will be needed to make the proposed method robust to noise and address the lack of fault data before it can be applied to actual equipment. The proposed studies will focus on lightweight AI models that can perform well in various fault diagnosis fields and be applied to actual industry applications at low power and cost.

## Figures and Tables

**Figure 1 sensors-23-03157-f001:**
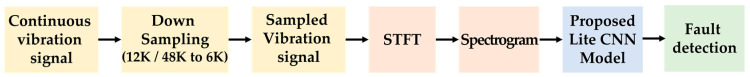
Bearing fault diagnosis process using the proposed lite CNN model.

**Figure 2 sensors-23-03157-f002:**
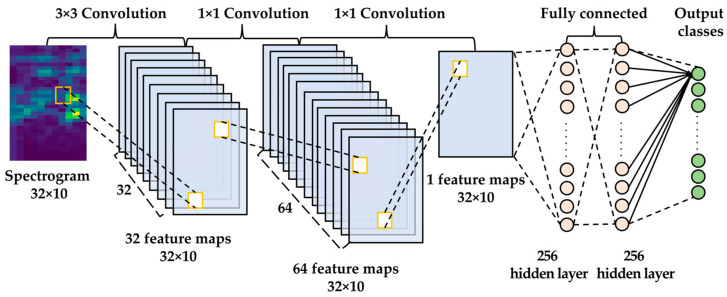
Architecture of the proposed lite CNN model for bearing fault diagnosis.

**Figure 3 sensors-23-03157-f003:**
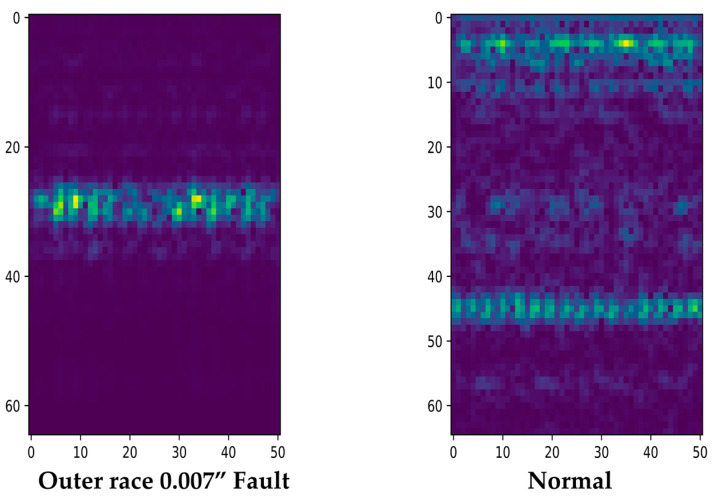
Example Spectrograms of Bearing Damage in Normal and Outer Direction Conditions.

**Figure 4 sensors-23-03157-f004:**
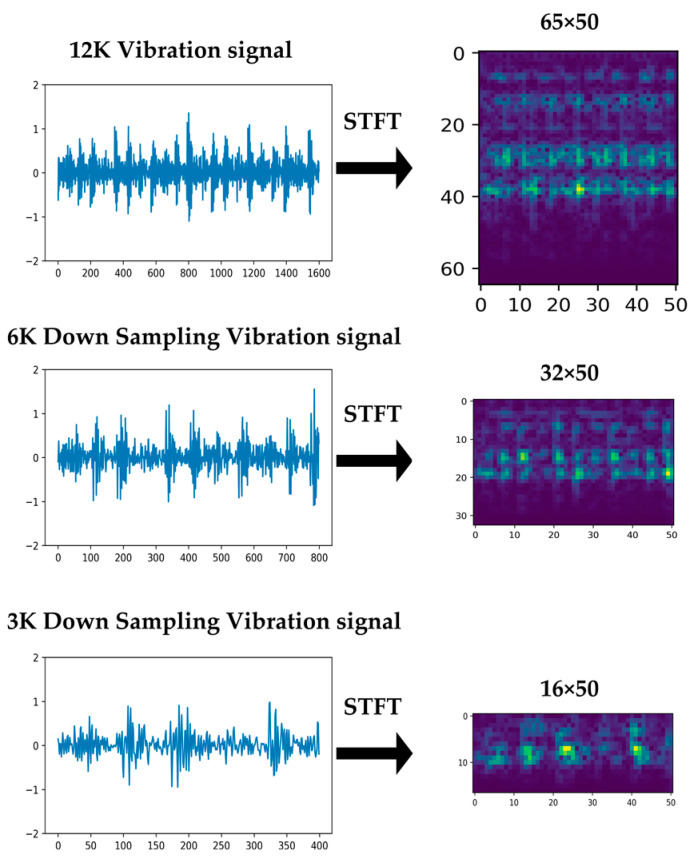
Downsampled vibration sensor signals and generated spectrograms.

**Figure 5 sensors-23-03157-f005:**
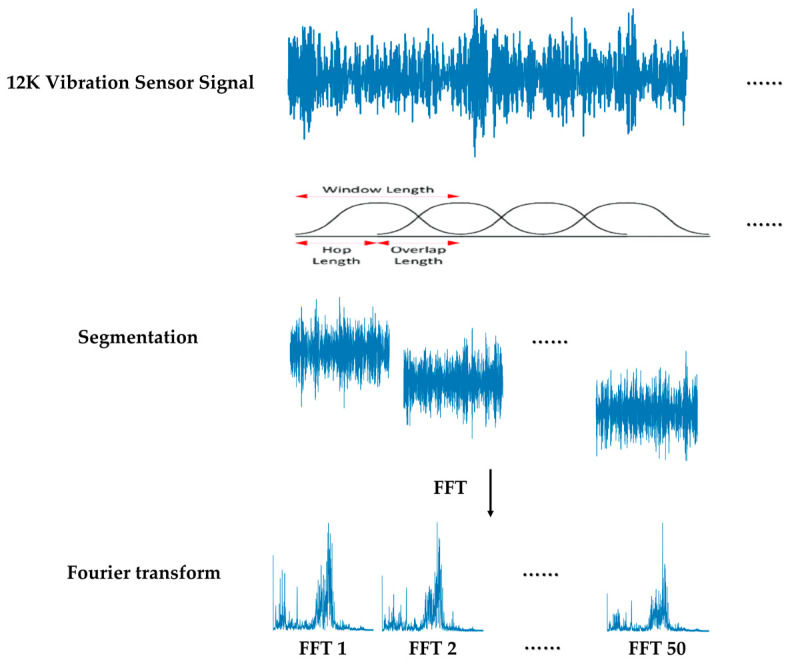
STFT process for generating spectrogram from vibration sensor signal [24].

**Figure 6 sensors-23-03157-f006:**
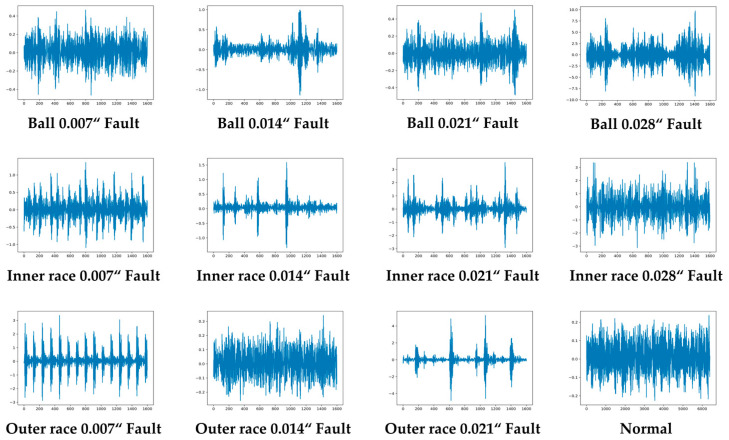
Samples of raw vibration sensor signals corresponding to each class.

**Figure 7 sensors-23-03157-f007:**
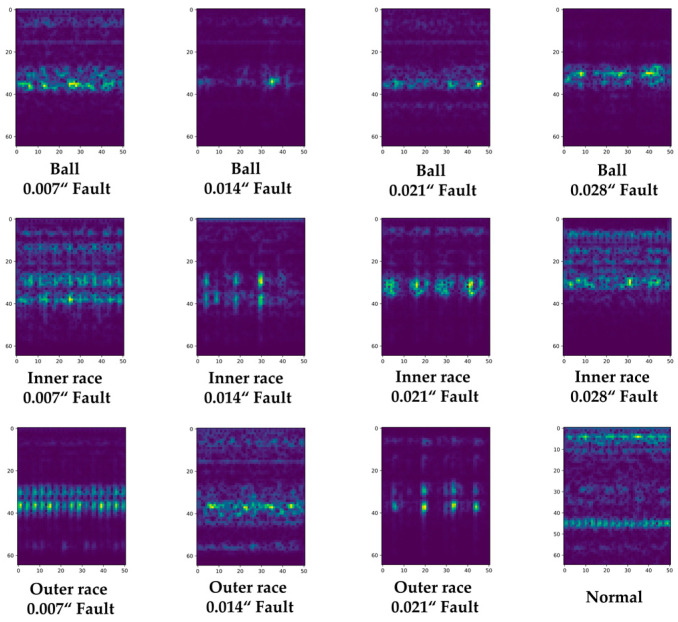
Generated Spectrogram corresponding to the raw vibration signal of each class shown in Figure 6.

**Figure 8 sensors-23-03157-f008:**
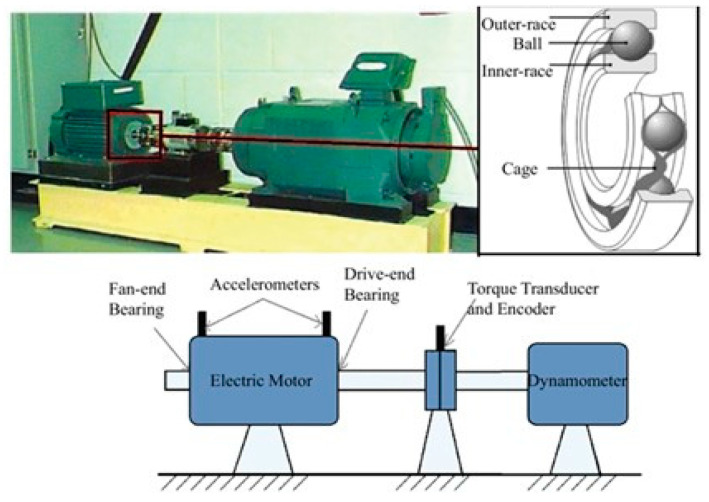
Bearing elements and an experimental setup of the CWRU bearing test rig for a ball bearing system [1].

**Figure 9 sensors-23-03157-f009:**
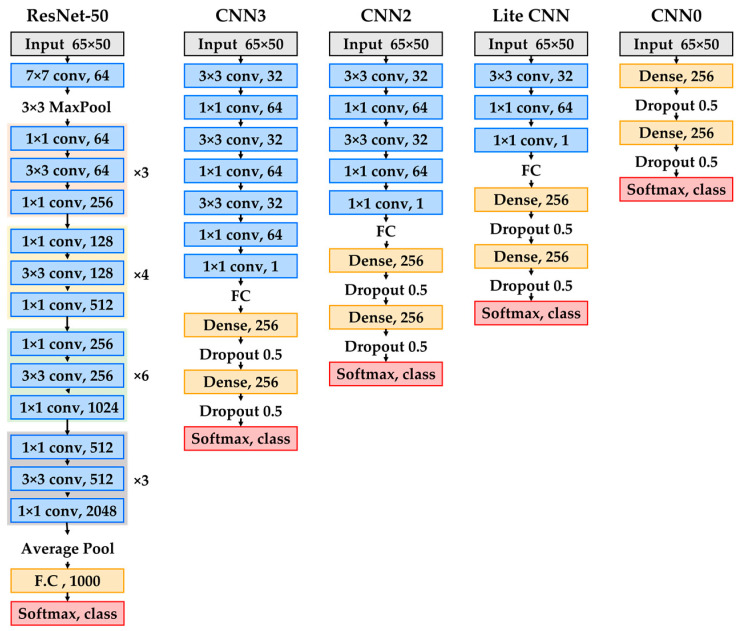
Proposed model (lite CNN) and comparison models (ResNet50, CNN3, CNN2, and CNN0).

**Figure 10 sensors-23-03157-f010:**
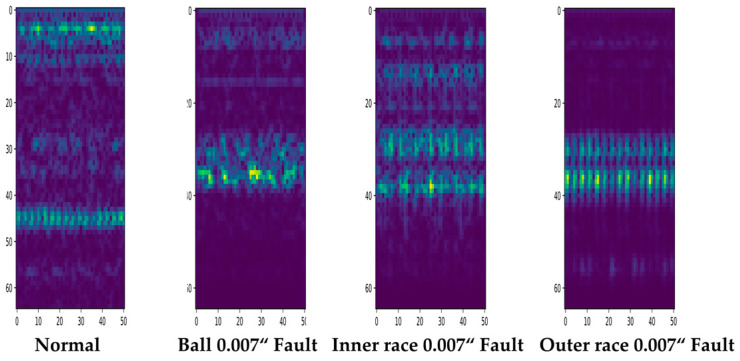
Examples of input spectrogram data of 65 × 50 size.

**Figure 11 sensors-23-03157-f011:**
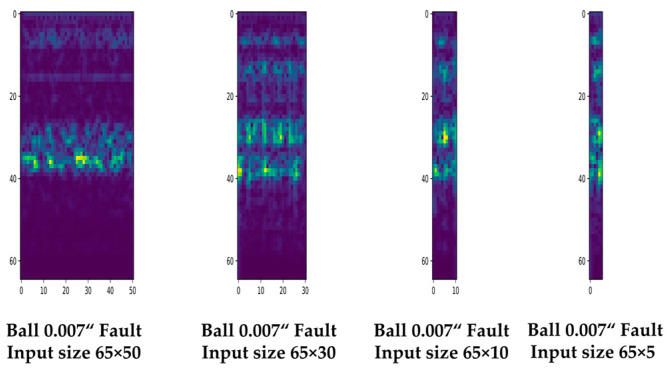
Examples of input spectrogram data of 65 × 50, 65 × 30, 65 × 10, and 65 × 5 sizes.

**Figure 12 sensors-23-03157-f012:**
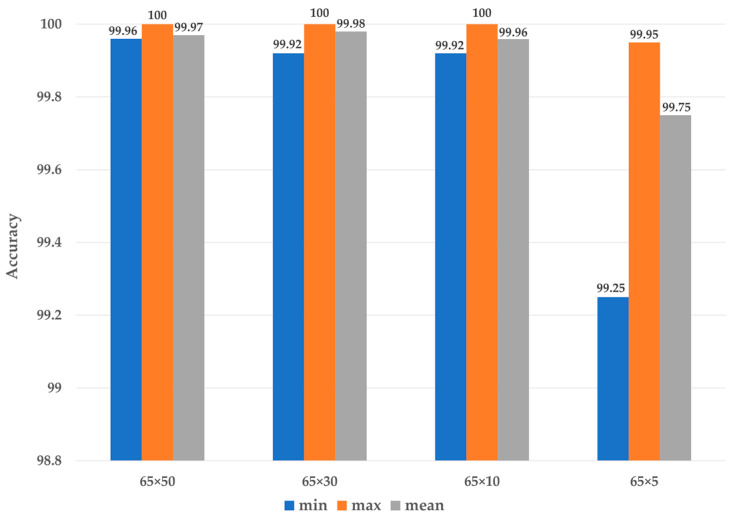
Comparison of the bearing fault diagnosis accuracy of lite CNN models with different input data sizes according to each different length of signal interval.

**Figure 13 sensors-23-03157-f013:**
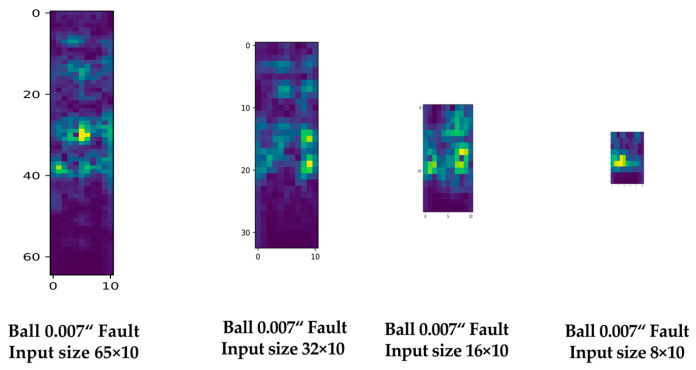
Examples of input spectrogram data of 65 × 10, 32 × 10, 16 × 10, and 8 × 10 sizes.

**Figure 14 sensors-23-03157-f014:**
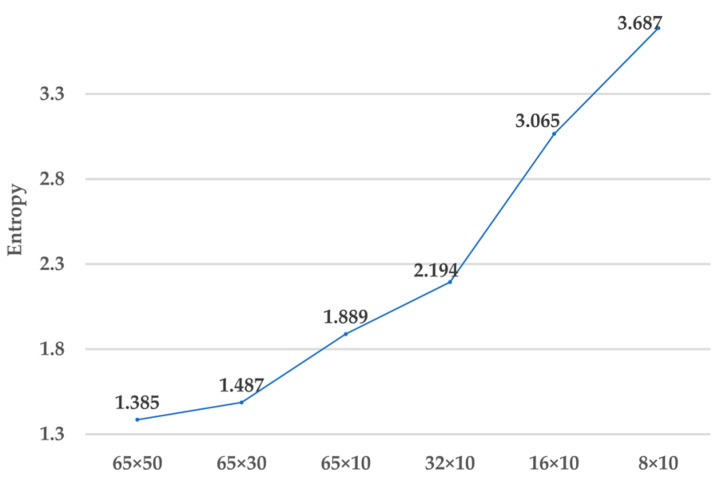
Comparison of entropy of spectrograms with different sizes.

**Table 1 sensors-23-03157-t001:** Parameters for each spectrogram with different sizes and corresponding time intervals.

Sample Rate	12K	48K
Spectrogram Size	65 × 50	
Time	0.1413 s	
Segmentation Size	1600	6400
Window Size	128	512
Hop Size	32	128

**Table 2 sensors-23-03157-t002:** Comparison of the bearing fault diagnosis accuracy performance and computational complexity measures (computation time, number of parameters, and FLOPs) of models using 65 × 50 size input data.

Model	Accuracy				Computation Time (s)			# of Parameters (K)		FLOPs (G)
	Min	Max	Mean	Std	Train	Predict	Total	Total	Dense	
ResNet50	99.92	100	99.97	0.036	147.393	1.307	148.700	47,187	23,606	0.718
CNN3	99.87	100	99.94	0.055	34.234	0.243	34.477	944	901	0.285
CNN2	99.75	100	99.95	0.07	27.544	0.24	27.785	924	901	0.151
Lite CNN	99.96	100	99.97	0.026	21.008	0.144	21.152	903	901	0.0177
CNN0	96.46	100	98.66	1.315	12.981	0.112	13.094	901	901	0.0018

**Table 3 sensors-23-03157-t003:** Parameters for each spectrogram with different sizes corresponding to the time intervals.

Spectrogram Size	65 × 50	65 × 30	65 × 10	65 × 5
Time	0.1413 s	0.0880 s	0.0346 s	0.0213 s
Segmentation Size	1600(6400)	960(3840)	320(1280)	160(640)
Window Size	128(512)	128(512)	128(512)	128(512)
Hop Size	32(128)	32(128)	32(128)	32(128)

**Table 4 sensors-23-03157-t004:** Comparison of bearing fault diagnosis accuracy, performance, and computation complexity measures (computation time, number of parameters, and Flops) of models using each different input data size.

Input Data Size	Accuracy				Computation Time (s)			# of Parameters (K)		FLOPs (G)
	Min	Max	Mean	Std	Train	predict	Total	Total	Dense	
65 × 50	99.96	100	99.97	0.027	21.008	0.144	21.152	903	901	0.0177
65 × 30	99.92	100	99.98	0.027	20.391	0.134	20.525	570	568	0.0107
65 × 10	99.92	100	99.96	0.031	19.632	0.120	19.752	238	235	0.00365
65 × 5	99.25	99.95	99.75	0.187	19.605	0.120	19.725	154	152	0.0019

**Table 5 sensors-23-03157-t005:** Parameters for each spectrogram with different sizes corresponding to different downsampling rates.

Spectrogram Size	65 × 10	32 × 10	16 × 10	8 × 10
Sample rate	12K	6K	3K	1.5K
Time	0.0346 s	0.0346 s	0.0346 s	0.0346 s
Segmentation Size	320(1280)	160(640)	80(320)	40(160)
Window Size	128(512)	64(256)	32(128)	16(64)
Hop Size	32(128)	16(64)	8(32)	4(16)

**Table 6 sensors-23-03157-t006:** Comparison of bearing fault diagnosis accuracy performance and computation complexity measures (computation time, number of parameters, and Flops) of lite CNN models using each input data size according to each corresponding sample rate.

Input Data Size	Accuracy				Computation Time (s)			# of Parameter (K)		FLOPs (G)
	Min	Max	Mean	Std	Train	Predict	Total	Total	Dense	
65 × 10	99.92	100	99.96	0.031	19.632	0.120	19.752	238	235	0.00365
32 × 10	99.70	100	99.86	0.112	18.694	0.120	18.814	153	151	0.00187
16 × 10	93.42	97.00	95.26	0.980	18.480	0.118	18.598	112	110	0.00100
8 × 10	83.54	87.21	85.20	1.142	18.395	0.119	18.514	92	89	0.00057

**Table 7 sensors-23-03157-t007:** Comparison of the bearing fault diagnosis accuracy performance and computation complexity measures (computation time, number of parameters, and Flops) of the proposed lite CNN model against ResNet based SOTA model.

Model	Accuracy			Computation Time(s)	FLOPs (G)	# of Parameters (K)
	Min	Max	Mean			
SOTA model (ResNet50 based on transfer learning)	99.90	100	99.95	294	3.8 over	23,900 over
Proposed lite CNN	99.92	100	99.95	18.326	0.00187	153

**Table 8 sensors-23-03157-t008:** CWRU Dataset Configuration.

Data Set	Fault Location	Sensor Location	# Classes
Set1	DE	DE	12
Set2		FE	10
Set3		BA	9
Set4	FE	DE	10
Set5		FE	10
Set6		BA	9
Set7	DE	DE	10
Set8		FE	10

**Table 9 sensors-23-03157-t009:** Bearing fault diagnosis accuracy performance and computation time of the proposed lite CNN model for all datasets (Set1~Set8).

Data	Accuracy				Computation Time (s)		
	Min	Max	Mean	Std	Train	Predict	Total
Set1	99.70	100	99.86	0.111539	18.694	0.120	18.814
Set2	99.85	100	99.93	0.050990	17.684	0.138	17.821
Set3	99.72	100	99.89	0.090752	16.054	0.133	16.187
Set4	99.65	99.85	99.76	0.056789	17.564	0.134	17.698
Set5	99.70	99.90	99.78	0.050990	17.596	0.139	17.735
Set6	99.72	99.94	99.81	0.066068	16.09	0.133	16.223
Set7	99.90	100	99.95	0.031623	17.58	0.138	17.718
Set8	99.95	100	99.98	0.025000	17.758	0.139	17.897

## Data Availability

The data presented in this study are available upon request from the corresponding author.

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
