# Peer review of "Lite and Efficient Deep Learning Model for Bearing Fault Diagnosis Using the CWRU Dataset"

_sensors, 2023, doi:10.3390/s23063157_

Round 1
Reviewer 1 Report
Following are few observations that would be needed to address:
1. A list of Abbreviations, Nomenclature, and their meaning should be needed. Use of abbreviations in a proper way.
2. Use a high-resolution image for Figure 03.[Right Side] and Figure 07.
3. If possible, the text needs an English wording revision in a few places, then corrected in the manuscript.
4. The conclusion should include some words of improvement, such as the increasing accuracy ratio. Please add discussion content indicating future research trends.
5. If a digital object identifier (DOI) is available, it should be included at the end of the reference.
6. Lite CNNs can offer advantages such as speed, portability, and good performance for certain tasks, but may have limitations in terms of complexity, generality, and interpretability. As with any model, the suitability of a Lite CNN for fault detection of rolling element bearings will depend on the specific use case and data. “How to quantitatively evaluate the performance”
7. Spectrogram entropy analysis has some limitations and drawbacks these are Sensitivity to signal length and segment size, Limited frequency resolution, Sensitivity to noise, Difficulty in interpreting results, and Lack of standardization. “ How this research paper addresses this limitation”
- Discuss the limitations and future directions: While the proposed model shows promising results, it is important to discuss the research's limitations and potential future directions. For example, the proposed model may not work well for other types of bearings or under different operating conditions. Identifying the limitations and suggesting potential directions for future research would make the paper more informative and impactful.
Reviewer 2 Report
This paper presents a lite deep learning model for bearing fault diagnosis. The DL model used consists of low complexity CNN model. The input of this model has low dimensionality by applying a minimal length of vibration signal in generating spectrograms and also down sampling of the vibration signal.
The results obtained are very satisfactory, and I believe that the manuscript has the potential for publication. However, some aspects need to be addressed, as detailed below.
Majors comments.
a. Highlight the novelty of the paper and better position it.
b. Summarize and reorganize the paper. The current length is too discouraging in regard to the claimed novelties.
c. Present the dataset (even quickly) in the introduction before presenting and commenting on the results you obtain with it.
d. Make the focus on the sensoring used in your research in order to improve the link with the journal scopes.
Minors comments.
a. Add the word Lite in the abstract (instead of Light) to be consistent with the title.
b. This sentence is not clear: “In order to make more light model, smaller input data has been considered by applying a minimal length of vibration signal in generating a spectrogram and also down sampling of vibration signal”. Please consider rewriting it.
c. Explain the acronym the first time it’s mentioned. For example, STFT on page 4.
d. Explain this statement: “In this paper, by examining the characteristics of the vibration sensor signal for bearing failure, the optimal input data dimension with low entropy that maintains the spatial characteristics of frequency for bearing failure is inferred.”
Round 2
Reviewer 2 Report
The authors have responded well to my remarks and have applied my recommendations. Therefore, I suggest the acceptance of the paper.